# Optimization of Vero Cells Grown on a Polymer Fiber Carrier in a Disposable Bioreactor for Inactivated Coxsackievirus A16 Vaccine Development

**DOI:** 10.3390/vaccines9060613

**Published:** 2021-06-07

**Authors:** Keda Chen, Chaonan Li, Ying Wang, Zhenwei Shen, Yikai Guo, Xiaoping Li, Yanjun Zhang

**Affiliations:** 1Shulan International Medical College, Zhejiang Shuren University, Hangzhou 310015, China; chenkd@zjsru.edu.cn (K.C.); 201930001207@stu.zjsru.edu.cn (C.L.); Ywang@Zjsru.edu.cn (Y.W.); shenlin7818@126.com (Z.S.); 201930001235@stu.zjsru.edu.cn (Y.G.); li-xp@zjsru.edu.cn (X.L.); 2Department of Virus Inspection, Zhejiang Provincial Center for Disease Control and Prevention, Hangzhou 310051, China

**Keywords:** coxsackie virus A16, polymer fiber paper carriers, disposable mini bioreactor, vaccine

## Abstract

At present, there are no vaccines available for hand, foot, and mouth disease, which is caused by Coxsackie virus A16 (CVA16) infection. In the present study, we isolated epidemic strains of CVA16 and optimized the production of the virus in Vero cells. The system comprised growing the infected cells on polymer fiber paper carriers in a serum-free medium containing 0.5% (*w*/*v*) lactalbumin hydrolysate a mini bioreactor. Disposable Bioflo310 and AmProtein Current perfusion bioreactors were used to monitor virus infection and Vero cell culture. The total number of cells increased from 1.5 × 10^9^ to 3.0 × 10^10^. In our optimized culture process, the virus titer reached 7.8 × 10^7^ TCID_50_/mL at three days after infection. The inactivated CVA16 prepared from our optimized culture procedure elicited a slightly higher neutralizing antibody titer compared with that derived from routine culture procedures. These results will promote the large-scale production of inactivated CVA16 vaccines using nonwoven polymer fiber paper cell cultures.

## 1. Introduction

With the emergence of new infectious diseases in recent years, the technology of the large-scale culture of viruses has displayed important advantages, which is of great significance for the development of vaccines an antibodies. Enterovirus 71 (EV71) and coxsackie virus A16 (CVA16) are the two main pathogens of hand, foot, and mouth disease (HFMD), which had spread recently in the Asia-Pacific region [1,2,3,4,5,6,7,8,9,10]. However, no CVA16 vaccine for HFMD is commercially available [10,11,12,13,14,15,16,17,18,19]. Increasing evidence has shown that both EV71 virus and CVA16 virus infections could cause serious neurological complications and even death. Therefore, when developing vaccines for HFMD, EV71 and CVA16 should be targeted [7,17,18,19,20,21,22,23].

Bioreactors have several advantages over cell factories and drum-type industrial vaccine production methods [24,25]. Fiber carriers are superior to other microcarriers for viral culture because of their low shear rate and friction force, particularly for viruses that can induce severe cytopathological effects [26,27,28].

Supplementation with serum during viral production might be associated with certain limitations, including high cost, risks of contamination with other viruses, residual bovine serum albumin (BSA) content, and difficulties related to subsequent purification [29,30,31]. Unfortunately, in large-scale bioreactor systems, the cost of serum-free media is very high. To solve these problems, the present study developed a relatively serum-free culture system via a process that uses lactalbumin hydrolysate (LH) as a medium supplement during the harvest period.

In the present study, we used the AmProtein Current Perfusion Bioreactor (ACPB) system from AmProtein (Hangzhou, China) in 2009 [27] to create a cell/virus bank and optimized the culture conditions for the production of viruses. The optimized conditions were then used in the ACPB and Bioflo310 bioreactors for CVA16 virus production [28].

## 2. Materials and Methods

### 2.1. Cell Lines and the Generation of the Cell Bank

Vero cells (derived from green monkey kidney cells; CCL81; ATCC, Manassas, VA, USA) were cultured in Dulbecco’s Modified Eagle Medium (DMEM) with high glucose (4.5 g/L) and 10% fetal bovine serum (FBS; GIBCO12800-082, Gibco, Grand Island, NY, USA) contents. The cells were grown in an incubator at 37 °C with 5% CO_2_, and they were passaged twice a week in T-75 tissue culture flasks. The cells were then added into a microbioreactor container and further expanded. A master cell bank (140 vials, 8 × 10^6^ cells/mL, 1.5 mL/vial) was generated for further use.

### 2.2. Plaque Purification and Virus Seeds Generation

The Coxsackie virus A16 epidemic strain (CVA16-2012ZJHFMD) was obtained from the Virus Laboratory Department of Zhejiang Provincial Center for Disease Control and Prevention. First, CVA16-2012ZJHFMD was grown in Vero cells in DMEM containing 10% FBS, and the plaques were purified. One of the five purified plaques was chosen as the pre-virus seed for subsequent amplification; the remainder were stored at 70 °C. The master seeds (160 vials, 6.0 × 10^7^ TCID_50_/mL, 1 mL/vial; TCID_50_, 50% tissue culture infectious dose) were stored at 70 °C. The cytopathic effect (CPE) was observed using microscopy, immunofluorescence (using antibody MAB979, 1:1000, Chemicon at Millipore (Billerica, MA, USA)), and TEM (Hitachi H-7650, Hitachi, Tokyo, Japan).

### 2.3. Determination of the Viral Titer and Residual BSA in the Harvest Supernatant

The TCID_50_ was used to determine the virus titer. In a 96-well plate, a Vero cell monolayer was inoculated with 10-fold serial dilutions of virus samples (from 10-1 to 10-9). A typical CPE of virus growth was observed under a microscope. The Reed and Muench method [32] was used to calculate the TCID_50_ values. A standard ELISA was used to detect the residual BSA (BSA-ELISA kit; Cygnus Technologies, Southport, NC, USA).

### 2.4. Vero Cell Culture in the Microbioreactor

The ACPB microbioreactor system comprises a container filled with a polymer fiber carrier (0.6 g) and a spin incubator (Appendix A). Vero cells (1.5 × 10^7^ cells) in 30 mL of culture medium were inoculated into 15 microbioreactor containers and cultured in a rotating incubator at 50 rpm at 37 °C with 5% CO_2_. A glucose test kit (Sigma, St. Louis, MO, USA) was used to detect the glucose concentration every 12 h. Crystal violet staining was used to determine the cell number. We analyzed the relationship between the glucose consumption rate (GCR) and the cell density. Vero cells at different densities (0.5 × 10^7^, 1.0 × 10^7^, 1.5 × 10^7^, and 2.0 × 10^7^/0.6 g) were added into the microbioreactor to assess the optimum seeding cell density. Every 24 h, according to the requirements for glucose consumption in high-density culture, we changed the medium.

### 2.5. Micro Bioreactor Optimization of the Yield of the Virus

Vero cells were cultured in the microbioreactor vessel and infected with CVA16 at different multiplicities of infection (MOIs) (MOI = 0.01, 0, 1, 0.5, and 1.0) to assess the effect of varying amounts of virus on the final virus yield. The temperature-dependence of virus yield was assessed by culturing CVA16-infected Vero cells in the microbioreactor vessel at different temperatures (31, 32, 33, 34, and 35 °C). We set three replicates for each group of cells, and the average was calculated. Samples were taken from the microbioreactor container at 24-h intervals during the cell growth process, and they were retained at 70 °C.

### 2.6. CVA16 Production Using Different Strategies of Medium Supplementation

Five media supplantation strategies were designed to identify the most practical virus harvesting system, which included detecting the effects of virus production serum-free medium (VP-SFM) (GIBCO) and lactalbumin hydrolysate (LH, GIBCO). In Group 1 (G1), CVA16 was inoculated into Vero cells in DMEM for 2 h, followed by supplementation using 10% FBS. The medium was changed every 24 h. The supernatant was collected every 24 h from 48 to 120 h after infection. In Group 2 (G2), CVA16 was inoculated into Vero cells in DMEM for 2 h, followed by the addition of VP-SFM. The medium was changed every 24 h. The supernatant was collected every 24 h from 48 to 120 h after infection. In Group 3 (G3), CVA16 was inoculated into Vero cells in DMEM for 2 h, followed by supplementation with glucose (2 g/L). The medium was changed every 24 h. The supernatant was collected every 24 h from 48 to 120 h after infection. In Group 4 (G4), CVA16 was inoculated into Vero cells in DMEM for 2 h, followed by supplementation with 10% FBS and incubation for 24 h. The medium was then replaced with DMEM containing glucose (2 g/L). The medium was changed every 24 h. The supernatant was collected every 24 h from 48 to 120 h after infection. In Group 5 (G5), CVA16 was inoculated into Vero cells in DMEM for 2 h, followed by supplementation with 10% FBS and incubation for 24 h. The medium was then replaced with DMEM containing various concentrations of LH (0.1, 0.25, 0.5, and 1% (*w*/*v*)) and glucose (2 g/L). The medium was changed every 24 h. The supernatant was collected every 24 h from 48 to 120 h after infection. Three repetitions were set for each group to calculate the mean virus titer.

### 2.7. Cell Expansion and Bioreactor Inoculation

Ten passages of Vero cells cultured in T75 flasks were performed, during which the doubling time and cell density remained stable. Only T75 flasks and microbioreactors were used in the seed cultivation process, which was more efficient and convenient compared with the use of a cell factory or drum. Fifteen T75 vials (containing 1.5 × 10^8^ or 1.0 × 10^7^ cells per vial) were transferred to 10 microbioreactors. After five days of culture, the micro bioreactor produced 1.5 × 10^9^ cells, which could be used as the seed for the bioreactor.

### 2.8. ACPB and NBS-Bioflo310 Culture

The polymer fiber carrier (150 g) was preloaded into the ACPB perfusion column and washed using 5 L of phosphate-buffered saline (PBS; 20 mM) overnight. The PBS was then replaced with cell growth medium. The Vero cells from the microbioreactor were added to the ACPB perfusion column, followed by a 2-h static incubation. During the cell culture period, the temperature was 37 °C, the vibration rate was 55 rpm, the pH ranged from 7.2 to 7.6, and the dissolved oxygen (DO) was 50%. In the first 24 h, a circulation rate of 100 mL/min was used, and then it was increased to 150 mL/min for the remaining period of cell culture. Culture continued for 6 d, after which we removed the culture medium and washed the ACPB three times using 5 L of PBS. Then, CVA16 (MOI = 0.01) in 10 L of fresh DMEM was pumped into the ACPB and statically incubated for 2 h. During the post-infection period, the temperature was 33 °C, the vibration speed was 50 rpm, the pH ranged from 7.2 to 7.4, and the DO was 60%. A circulation rate of 150 mL/min was maintained. Throughout the process, every 24 h, 5 L of medium were replaced. During the post-infection period, we harvested 5 L of supernatant at 48, 72, and 96 h post-infection (hpi). At 120 hpi, we harvested 10 L of supernatant. The infected cells were kept on the polymer fiber carrier, which contained 5 L of PBS, and processed through three freeze–thaw cycles. An illustration of the ACPB is presented in Appendix A. The NBS-Bioflo310 bioreactor (New Brunswick Scientific (NBS), Edison, NJ, USA) is similar to the ACPB, except that it uses a nonwoven polymer fiber paper as the cell carrier and has some differences in its structure (non-disposable) and working procedures. The working volume of the Bioflo310 is 4.5 L. When using the Bioflo310, we changed 4.5 L of medium every 24 h. After CVA16 infection, we harvested 4.5 L of supernatant at 48, 72, 96, and 120 hpi (a total of 18 L). The infected cells remained on the polymer fiber carrier, which was filled with 4 L of PBS and subjected to three freeze–thaw cycles. To produce CVA16, the parameters and culture strategies were the same for the NBS-Bioflo310 and ACPB bioreactors.

### 2.9. Inactivated CVA16 Preparation and Immunization of Mice

Samples of virus harvested from the bioreactors were purified using sucrose gradient ultracentrifugation (as described previously [33]), mixed with formaldehyde at a ratio of 4000:1, and then incubated for 30 days at 4 °C. The inactivated CVA16 particles (10 μg/100 μL) were absorbed onto aluminum phosphate for 3 h before use. We purchased female BALB/c mice (6–8 weeks old) from the Shanghai Laboratory Animal Center, China. The Guide for the Care and Use of Laboratory Animals of Zhejiang Province was followed during all animal experiments, and the local ethics committee approved the study. The alum-absorbed inactivated CVA16 (10 μg/100 μL/mouse) was used for the intraperitoneal immunization of mice, and the same dose was used to boost the mice after two weeks. One week after the booster immunization, the immunized mice were bled, and serum samples were tested using a neutralizing assay. CVA16 serum neutralizing antibody titers were measured using standard protocols [34]. The starting dilution was eight-fold, followed by a two-fold serial dilution. The cutoff level for seropositivity was 8.

## 3. Results

### 3.1. GCR Determination in the Mini Bioreactor

Every 24 h, the Vero cell culture GCR was measured in the microbioreactor. Figure 1A shows a schematic diagram of the GCR and cell growth of Vero cells in the microbioreactor culture vessel. After five days of culture, the total number of cells increased from 1.5 × 10^7^ to 1.5 ± 0.2 × 10^8^. There was a direct relationship between the GCR and cell growth. During 96–120 h of culture in the 30 mL system, the maximum GCR was 3.36 g/L/24 h. The cell numbers and the GCR gradually decreased after 120 h because the maximum capacity of the fiber carriers had been reached and the frequency of medium change might not have been sufficient for high-density cells.

Using different cell seeding densities shifted the time taken to reach the maximum GCR in the mini bioreactor vessels (Figure 1B). Vero cells at 1.5 × 10^7^ cells/0.6 g was identified as the optimal seeding density, because the GCR was similar to that when using 2.0 × 10^7^ cells, but the maximum GCR was reached earlier compared with using the other two seeding densities.

### 3.2. Infection Parameter Optimization in the Microbioreactors

According to our previous results, we used a seeding density of 1.5 × 10^7^ cells/0.6 g plate and the optimal culture time of five days. After inoculation with CVA16 virus, the supernatant was collected every 24 h and the virus titer was determined. When the MOI was 0.5, the increase of virus titer reached its peak at 96 hpi. The same trend was observed when the MOI was 0.1 and 1; however, the virus titer was lower when the MOI was 0.1 (Figure 1C). Consequently, the optimal MOI for virus production was 0.5. Compared with the other groups, we found that the virus titer was higher when the bioreactors were incubated at 32 and 33 °C. At 33 °C, a total harvest titer of 1.0 × 10^8^ TCID_50_/mL was achieved (Figure 1D).

### 3.3. Dependence of CVA16 Viral Yield on Different Culture Strategies and Medium Supplements

Five different strategies for virus production were used: G1 (FBS), G2 (VP-SFM), G3, G4, and G5 (LH at 0.1%, 0.25%, 0.5% and 1% (*w*/*v*), respectively). CVA16 virus was inoculated in the microbioreactors at an MOI of 0.5 at 33 °C. The supernatant was collected at 48, 72, 96, and 120 hpi, and the virus titer was measured. The virus yields of G1, G2, and G5 (0.5% LH) were higher than those of the other groups at 8.0, 7.7, and 7.8 Log10TCID_50_/mL, respectively. There was no difference in virus yield between G5 (0.5% LH) and G5 (1% LH), suggesting that G5 (0.5% LH) could be the most suitable strategy for the production of CVA16 (Figure 2).

### 3.4. ACPB and NBS-Bioflo310 Culture

Ten microbioreactors containing 1.5 × 10^9^ Vero cells were used in the first ACPB culture experiment (ACPB-0). Static incubation for 2 h was used to facilitate cell attachment, during which about 97% of the cells attached. During culture, the glucose level was maintained above 1.5 mg/mL by changing 5 L of medium every 24 h. Six days later, the crystal violet staining of the ACPB fibrous carrier showed that the total number of Vero cells was 3.0 × 10^10^. The cell densities of different parts of the perfusion column were similar, confirming the even distribution of the cells (Figure 3A).

The second experiment (ACPB-1) used the G1 strategy, in which the highest virus titer was 8.0 Log10TCID_50_/mL (Figure 3B). Twenty-five liters of supernatant were harvested in total. The next three experiments (ACPB-2, 3, and 4) utilized the G5 (0.5% LH) strategy. The highest virus titer achieved was 7.9 Log10TCID_50_/mL (7.8 × 10^7^ TCID_50_/mL) (Figure 3C). Twenty-five liters of supernatant were harvested in total. The average virus titer was 7.5 × 10^7^ TCID50/mL, with a total yield of 1.88 × 10^12^ TCID50. The last test used the NBS-Bioflo310 bioreactor and the G5 (0.5% LH) strategy. The highest virus titer achieved was 7.9 Log10TCID50/mL (7.7 × 10^7^ TCID50/mL) (Figure 3D). Eighteen liters of supernatant were harvested in total. The average virus titer was 7.7 × 10^7^ TCID50/mL, with a total yield of 1.39 × 10^12^ TCID50. The total yield (TCID50) of the cell lysate (intracellular) and supernatant (extracellular) was calculated (Figure 4).

### 3.5. Formalin-Inactivated CVA16 Immunogenicity in Mice

To further investigate whether the different medium supplementation strategies affected the immunogenicity of CVA16, the viral stocks harvested from ACPB-1/G1, ACPB-2/G5, and Bioflo310/G5 at 96 hpi were inactivated using a formaldehyde solution and absorbed onto aluminum phosphate. BALB/c mice (three groups of six mice) were immunized with inactivated CVA16 virions on a two-week schedule. Inactivated CVA16 prepared from ACPB-2/G5 and Bioflo310/G5 induced a slightly higher neutralizing antibodies titer compared with that of ACPB-1/G1 (Figure 5).

## 4. Discussion

CVA16 is one of the most common viruses associated with hand, foot, and mouth disease in Asia. Moreover, CVA16 epidemics in Zhejiang province (China) from 2008 to 2012 caused more severe symptoms in children, such as neurological diseases and death. Further complicating the situation, EV71 virions do not elicit neutralizing antibodies that cross-react against CVA16 [35]. Therefore, it is necessary to develop a candidate CVA16 vaccine based on Vero cell whole virus production.

Chong et al. (2012) described a process of CVA16 production using cytodex1 as a microcarrier [33]. However, nonwoven polymer fiber carriers immobilized in cell cultures have a low shear rate and friction force, and they are now considered to be superior to other microcarriers (e.g., the cytodex series) for viral culture. During microbioreactor culture, the cells are inaccessible; therefore, we developed an indirect method to estimate cell density. We investigated whether the GCR could be used to indirectly indicate the estimated cell growth and density in mini bioreactors. The results showed that the GCR could reliably estimate the cell density and could be used to track cell growth in the model system.

In this study, we determined that for CVA16 production, the optimal MOI was 0.5 and the optimal temperature was 32 or 33 °C. The effects of pH and DO should be further investigated in a large-scale bioreactor [36,37]. To meet the requirement of GMP and the ability to practically manufacture a vaccine, we established standards to reduce the cost of production and increase the quality of the vaccine. The viral incubation time was limited to less than five days, and after CVA16 inoculation, we used different medium supplementation strategies. Moreover, only the supernatants (extracellular virus) were harvested, aiming to decrease the contamination of residual host cell proteins and DNA in the final product as much as possible. The intracellular virus from cell lysates were stored as virus seeds. To reduce the costs associated with the culture medium, we explored a suitable medium supplementation strategy. The results identified that G5 was the most suitable medium supplementation strategy. Using this strategy, the viral yield was negligible before 48 hpi; therefore, the supernatant was only harvested after this time point. In addition, the supernatant contained substantially less intracellular content, which might be convenient for downstream purification. Prior to viral infection, the cells were supplemented with FBS and had sufficient nutrition reserves. Following infection, the cells received FBS for the first 24 h and then received 5% LH for the remainder of the infection period. The final virus titer suggested that this strategy could supply sufficient nutrients for the cells over the next three days. Thus, as a medium supplement, LH is more economic and suitable than FBS or VP-SFM in this system.

Inactivated virus immunogenicity is a crucial factor that affects vaccine progression to clinic trials. The purity of the harvested CVA16 virus and the downstream purification methods can affect the immunogenicity of the inactivated CVA16 virus. The observed increase in neutralizing activity might be attributable to the use of a relatively “serum-free” strategy and the low shear force during the culture of virus.

COVID-19 pneumonia has spread worldwide. Many studies have found that the tissue phagocytosis and pathogenicity of SARS-CoV-2 is relatively similar to CVA16 in a Vero cell model [38,39]. Our results may constitute valuable information for the development of a large-scale cell culture process for producing an inactivated SARS-CoV-2 vaccine.

## 5. Conclusions

In conclusion, we developed a model system and suitable culture method for CVA16 production that might provide useful information to develop a CVA16 vaccine to resist HFMD infection.

## Figures and Tables

**Figure 1 vaccines-09-00613-f001:**
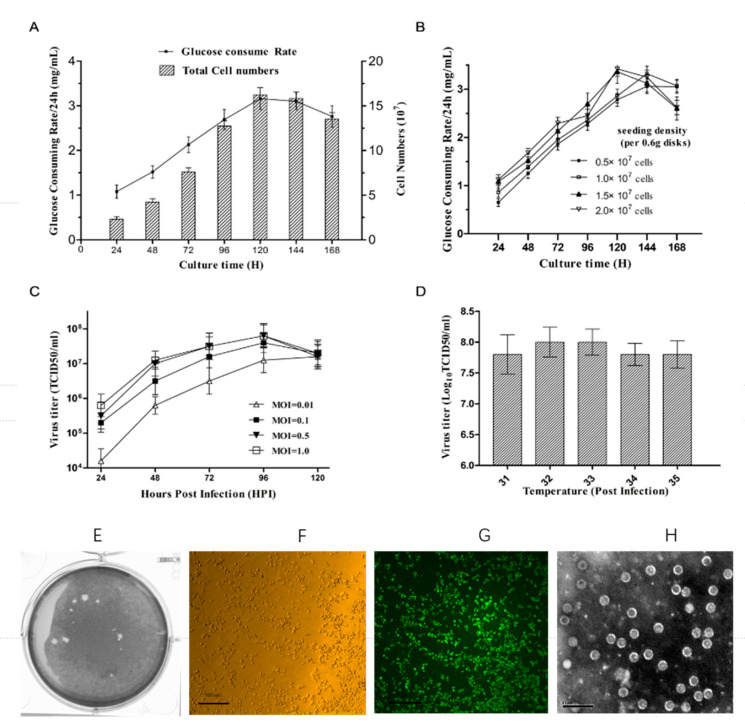
Vero cell culture in a micro bioreactor and optimization of the virus yield. (**A**) The total cell count in a container (volume = 30 mL;) and the GCR (black square; grams of glucose consumed every 24 h per liter) in the culture container of the micro bioreactor. (**B**) GCR in the bioreactor using inocula at different cell densities (0.5 × 10^7^ cells (black squares), 1.0 × 10^7^ cells (white squares), 1.5 × 10^7^ cells (black triangles), and 2.0 × 10^7^ cells (white triangles). (**C**) Viral production (yield) at different MOIs. MOI = 0.01 (white triangle), 0.1 (black square), 0.5 (black triangle), and 1.0 (white square). (**D**) Viral production under different temperatures (31, 32, 33, 34, and 35 °C) in the mini bioreactor. (**E**) Plaque purification of clonal CVA16. (**F**) CPE of CVA16-infected Vero cells at 68 hpi; magnification: 200. (**G**) Immunofluorescence staining of CVA16-infected Vero cells at 68 hpi; magnification: 200. (**H**) Transmission electron microscopy of CVA16. Scale bar represents 50 nm. GCR, glucose consumption rate; MOI, multiplicity of infection; CVA16, Coxsackie virus A16; CPE, cytopathic effect; hpi, hours post-infection.

**Figure 2 vaccines-09-00613-f002:**
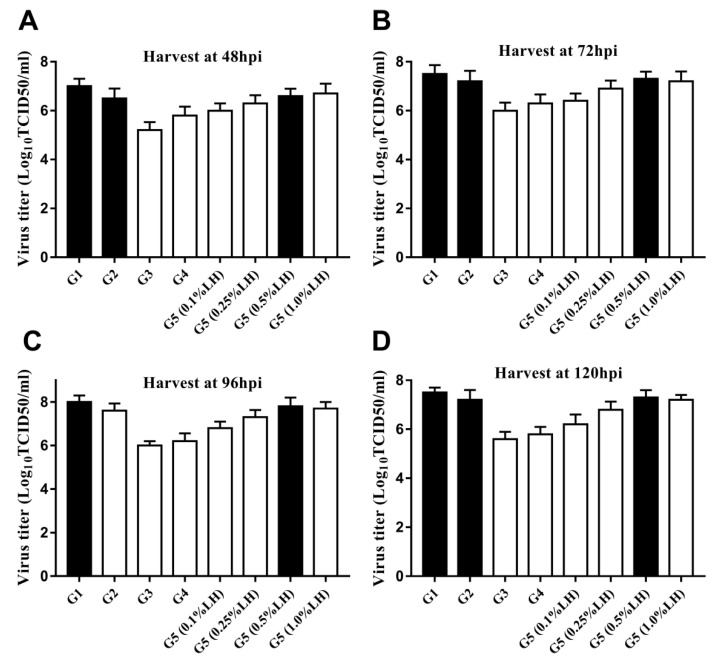
CVA16 virus production using different culture strategies and medium supplements. The virus titer of the supernatant was collected after growth using five different medium addition strategies: (**A**) 48 hpi, (**B**) 72 hpi, (**C**) 96 hpi, and (**D**) 120 hpi (group G1–G5). CVA16, Coxsackie virus A16; hpi, hours post-infection.

**Figure 3 vaccines-09-00613-f003:**
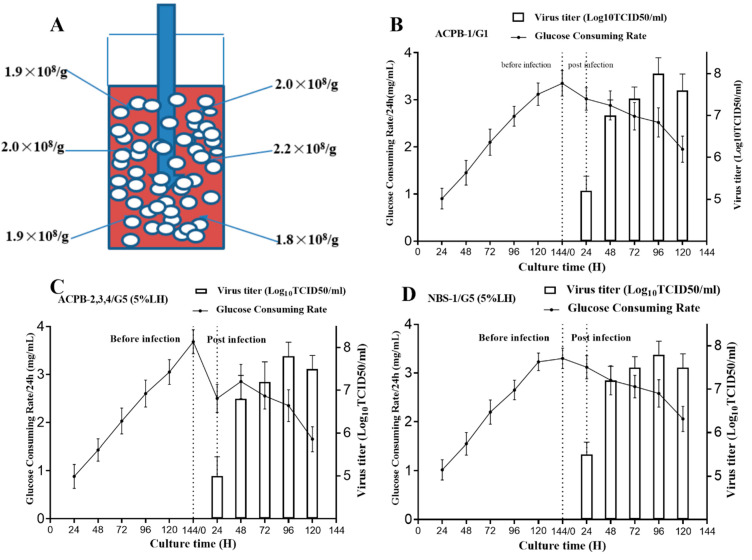
Culture in the NBS-Bioflo310 and ACPB bioreactors. (**A**) Distribution of cells in a 5 L perfusion column (5-L) containing 150 g of polymer fiber carriers in ACPB-0. (**B**) The viral titer and GCR in ACPB-1 (*n* = 3, 3 replicates in one batch) utilizing the G1 strategy. (**C**) The viral titer and GCR in ACPB-2, 3, and 4 (*n* = 9, 3 replicates × 3 batches) utilizing the G5 (5% LH) strategy. (**D**) The viral titer and GCR in NBS-1 (*n* = 3, 3 replicates in one batch) in the NBS-Bioflo310 system utilizing the G5 (5% LH) strategy. NBS-Bioflo310, New Brunswick Scientific Bioflo310 bioreactor; ACPB, AmProtein Current Perfusion Bioreactor; GCR, glucose consumption rate; LH, lactalbumin hydrolysate.

**Figure 4 vaccines-09-00613-f004:**
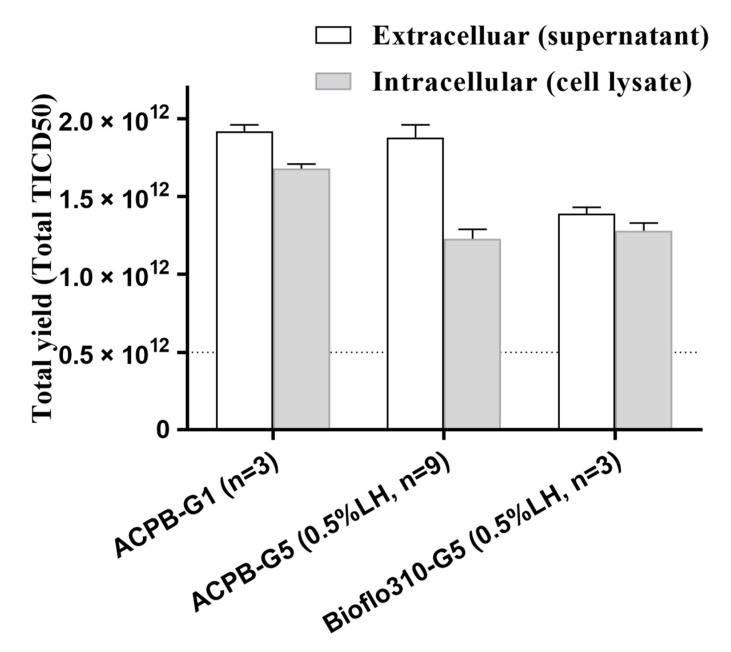
Total yield of virus in the cell lysates (intracellular) and supernatants (extracellular) from the ACPB and Bioflo310 systems using the G1 and G5 strategies. NBS-Bioflo310, New Brunswick Scientific Bioflo310 bioreactor; ACPB, AmProtein Current Perfusion Bioreactor. The determination of the residual BSA of the harvested supernatant from the ACPB and NBS-Bioflo310 bioreactors showed a substantial decrease when using the G5 (0.5% LH) strategy compared with using the G1 strategy (Table 1).

**Figure 5 vaccines-09-00613-f005:**
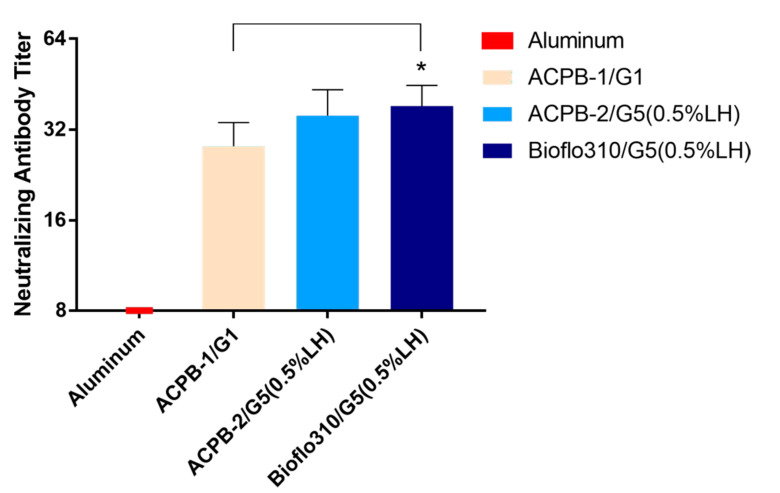
Neutralizing antibodies produced by mice immunized using the inactivated CVA16 vaccine. Titers of neutralizing antibodies (GMTs and 95% CI) in serum samples collected from mice immunized with the inactivated CVA16 vaccine or aluminum as a control (*n* = 8) at 14 days after the first immunization. The neutralization test was based on the highest dilution of the serum that stopped cells from undergoing a cytopathic effect. Data are presented as means, *n* = 8, * *p* < 0.05 compared with the ACPB-1/G1. CVA16, Coxsackie virus A16; GMT, geometric mean titer; CI; confidence interval.

**Table 1 vaccines-09-00613-t001:** Residual BSA of the harvested supernatants at 48, 72, 96, and 120 hpi from the ACPB and NBS-Bioflo310 bioreactors. BSA, bovine serum albumin; hpi, hours post-infection; ACPB, AmProtein Current perfusion bioreactor; NBS-Bioflo310; Bioflo310 from New Brunswick Scientific; LH, lactalbumin hydrolysate.

Source/Harvest Time	48 hpi	72 hpi	96 hpi	120 hpi
ACPB-1/G1	8.0 × 10^6^ ng/mL	7.6 × 10^6^ ng/mL	8.6 × 10^6^ ng/mL	8.8 × 10^6^ ng/mL
ACPB-2/G5 (0.5% LH)	280 ng/mL	86 ng/mL	16 ng/mL	5.6 ng/mL
Bioflo310/G5 (0.5% LH)	265 ng/mL	35 ng/mL	3.2 ng/mL	undetectable

## Data Availability

Data are contained within the article or supplementary material. The data presented in this study are available in [insert article or supplementary material here].

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
