# Peer review of "Optimization of Vero Cells Grown on a Polymer Fiber Carrier in a Disposable Bioreactor for Inactivated Coxsackievirus A16 Vaccine Development"

_vaccines, 2021, doi:10.3390/vaccines9060613_

Round 1

Reviewer 1 Report

The manuscript “Optimization of Vero cells grown on a polymer fiber carrier in a disposable bioreactor for inactivated 3 Coxsackievirus A16 vaccine development” by Chen et. al., is timely development in technique and will of broad interest. However, a vaccine against the CV-A16 virus was made using the chemically inactivated virus in Vero or KMB cells (Cai et. al., Vaccine, 2013). In this manuscript, it is noted that the use of bioreactors and the associated technique are superior because of the kind of carrier is used. This is a technical paper and does not show the efficacy of the vaccine against virus infection.

I have the following comments:

Major comments:

  1. Please discuss in detail why this technique is better than the Cai et. al., 2013 paper.
  2. An experiment showing the efficacy of the neutralizing antibody by infecting mice with the mouse-adapted strain CA16-MAV is recommended and this experiment would greatly increase the value of the manuscript.

Minor comments:

  1. In all the graphs the authors should show mean and standard deviation and the legend should have the number of replicates (n).
  2. Please add more relevant papers in the introduction.
  3. At the beginning of the result section, the instruction as to how to write the result shouldn’t be there.

Reviewer 2 Report

In this study, the authors have developed the optimized protocol for large scale production of inactivated coxsackie virus A16 (CVA16) vaccines. This study also described that inactivated CVA16 vaccines produced by using the optimized protocols successfully induce high levels of neutralizing antibodies against CVA16.
Overall, the paper is well organized. The experimental approach is sound, and the experiments were well executed. The results have been clearly presented and support the main conclusions presented by the authors. 

Specific Comments:
1) In figure 3 and 4, error bars are missing in ACPB-G1 and Bioflo310-G5 groups. It should be indicated how many biological replicates were included in each data set.
2) In figure 5, have you tested the protective efficacy in a challenge model of CVA16 lethal infection? Have you tested T cell responses? This information would be informative and improve this paper.

Round 2

Reviewer 1 Report

The revised manuscript by Chen et. al. answered most of my queries.  They have mentioned the research on the efficacy of the vaccine by this method is underway. Given the current situation with the COVID-19 pandemic, I believe, the manuscript will be helpful to be out and widely read by the interested scientific community. 

Author Response

Response to Reviewer 1 Comments

Point 1: The revised manuscript by Chen et. al. answered most of my queries.  They have mentioned the research on the efficacy of the vaccine by this method is underway. Given the current situation with the COVID-19 pandemic, I believe, the manuscript will be helpful to be out and widely read by the interested scientific community.

Response 1: Thanks for your understanding and approval of our study. We wil carry on the study and try our best to solve the real technique problem existing in vaccine development. This manuscript has been improved with your help and advice.  Thanks again!